# Evaluation of the Drug-Induced Liver Injury Potential of Saxagliptin through Reactive Metabolite Identification in Rats

**DOI:** 10.3390/pharmaceutics16010106

**Published:** 2024-01-13

**Authors:** Ki-Young Kim, Yeo-Jin Jeong, So-Young Park, Eun-Ji Park, Ji-Hyeon Jeon, Im-Sook Song, Kwang-Hyeon Liu

**Affiliations:** 1BK21 FOUR KNU Community-Based Intelligent Novel Drug Discovery Education Unit, Research Institute of Pharmaceutical Sciences, College of Pharmacy, Kyungpook National University, Daegu 41566, Republic of Korea; neanic12@naver.com (K.-Y.K.); duwls9902@gmail.com (Y.-J.J.); soyoung561@hanmail.net (S.-Y.P.); roseej98@naver.com (E.-J.P.); kei7016@naver.com (J.-H.J.); 2Mass Spectrometry Based Convergence Research Institute, Kyungpook National University, Daegu 41566, Republic of Korea

**Keywords:** drug-induced liver injury, glutathione, metabolism, saxagliptin, vildagliptin

## Abstract

A liver injury was recently reported for saxagliptin, which is a dipeptidyl peptidase-4 (DPP-4) inhibitor. However, the underlying mechanisms of saxagliptin-induced liver injury remain unknown. This study aimed to evaluate whether saxagliptin, a potent and selective DPP-4 inhibitor that is globally used for treating type 2 diabetes mellitus, binds to the nucleophiles in vitro. Four DPP-4 inhibitors, including vildagliptin, were evaluated for comparison. Only saxagliptin and vildagliptin, which both contain a cyanopyrrolidine group, quickly reacted with L-cysteine to enzyme-independently produce thiazolinic acid metabolites. This saxagliptin–cysteine adduct was also found in saxagliptin-administered male Sprague–Dawley rats. In addition, this study newly identified cysteinyl glycine conjugates of saxagliptin and 5-hydroxysaxagliptin. The observed metabolic pathways were hydroxylation and conjugation with cysteine, glutathione, sulfate, and glucuronide. In summary, we determined four new thiazoline-containing thiol metabolites (cysteine and cysteinylglycine conjugates of saxagliptin and 5-hydroxysaxagliptin) in saxagliptin-administered male rats. Our results reveal that saxagliptin can covalently bind to the thiol groups of cysteine residues of endogenous proteins in vivo, indicating the potential for saxagliptin to cause drug-induced liver injury.

## 1. Introduction

Drug-induced liver injury (DILI) has been the most prevalent cause of hepatic dysfunction, drug failures in clinical trials, and drug withdrawals from the pharmaceutical market over the past 50 years [1]. DILI has been associated with >900 marketed drugs, and drugs withdrawn for DILI include bromfenac, iproniazid, and troglitazone [2]. Idiosyncratic reactions cause most DILI cases. The irreversible covalent binding of a drug or its reactive metabolites to endogenous proteins cause idiosyncratic DILI, and drug–protein adducts acquire immunogenicity as antigens and illicit immunological responses [3,4,5]. Some drugs, such as aspirin and ampicillin, covalently bind to proteins [6]. Reactive intermediates produced by the cytochrome P450 enzyme directly bind to proteins for other drugs, such as acetaminophen [6]. Nucleophilic sites of endogenous proteins, such as the thiol residue of cysteine, are targets for drugs and their electrophilic intermediates. N-acetyl-L-cysteine, L-cysteine, or glutathione trap these reactive electrophiles. Cysteine and glutathione conjugates of drugs are considered markers of covalent binding with endogenous proteins [4,7].

Dipeptidyl peptidase-4 (DPP-4) inhibitors are promising medicines for treating type 2 diabetes mellitus (DM) and are expected to provide advantages over traditional DM treatments, including a low risk of hypoglycemia and weight gain [8]. However, vildagliptin-induced hepatic dysfunction has been reported in rare cases [9,10]. Anagliptin is known to produce metabolites with the same structure as vildagliptin, and these metabolites may contribute to liver injury [4,10]. Mizuno et al. revealed that the electrophilic nitrile moiety of vildagliptin is irreversibly converted to a thiazoline acid to form the vildagliptin–cysteine conjugate through a mechanism studied for elucidating vildagliptin-induced liver injury [3,4]. They revealed three thiazoline-containing thiol conjugates in vildagliptin-administered rats [3]. Additionally, anagliptin, which consists of a cyanopyrrolidine group as vildagliptin, was unstable on incubation with L-cysteine, whereas alogliptin, linagliptin, and sitagliptin, which have no cyanopyrrolidine group, were stable [4]. Such covalent binding of vildagliptin would occur in humans, and it may elicit unpredictable immunological responses in humans.

Saxagliptin is a potent and selective DPP-4 inhibitor that is globally used for treating type 2 DM. An oral dose of 2.5 or 5 mg saxagliptin once daily is recommended [11]. Saxagliptin is known to have nephroprotective, anti-oxidant, anti-inflammatory, and anti-apoptic effects [12]. Large clinical trials reported no instances of clinically clear liver injury. However, saxagliptin-induced liver injury has been reported in rare cases. Thalha et al. recently presented a case of hepatotoxicity caused by Kombiglyze (saxagliptin of 5 mg plus metformin of 1000 mg) in Malaysia [13]. The patient, showing palpable hepatomegaly and no stigmata of liver disease, demonstrated increased alanine aminotransferase (307 IU/L) and gamma-glutamyltransferase (808 IU/L) levels one week after the first Kombiglyze administration. He was diagnosed with steatohepatitis and marked intrahepatic cholestasis through a liver biopsy. The liver function returned to normal after withholding Kombiglyze.

Saxagliptin is rapidly absorbed after oral administration and is generally well tolerated [14]. The primary metabolic pathway of saxagliptin is adamantine moiety hydroxylation to form 5-hydroxysaxagliptin, which is a pharmacologically active metabolite in humans [8,15]. Cytochrome P450 3A4 and 3A5 mainly contribute to saxagliptin hydroxylation [8]. Additional minor metabolites from sulfation, glucuronidation, and nonenzymatic degradation have been determined [8]. However, glutathione-conjugated and related metabolites remained unidentified in animals or humans. Saxagliptin exhibits a cyanopyrrolidine moiety in its structure, similar to vildagliptin and anagliptin, indicating that cysteine or glutathione conjugates might be formed in vitro and in vivo, similar to vildagliptin. Analogous covalent binding to the thio group of endogenous proteins may initiate immune-mediated adverse reactions, causing hepatotoxicity. This study investigated whether thiol adducts of saxagliptin, such as cysteine and glutathione conjugates, form in rat liver microsomes and rats. These results will elucidate the potential for and mechanism of saxagliptin-induced liver injury.

## 2. Materials and Methods

### 2.1. Chemicals and Reagents

Saxagliptin, trelagliptin, and gemigliptin (Figure 1) were purchased from AdooQ Bioscience (>98%, Irvine, CA, USA); linagliptin and 5-hydroxysaxagliptin from Toronto Research Chemicals (Toronto, Canada); ammonium acetate (99.99%), β-nicotinamide adenine dinucleotide phosphate sodium salt hydrate (NADP^+^), dimethyl sulfoxide (DMSO, >99.7%), glucose-6-phosphate (G6P), glucose-6-phosphate dehydrogenase (G6PD), L-cysteine, magnesium chloride, reduced glutathione (GSH, ≥99%), 3′-phosphoadenosine-5′-phosphosulfate (PAPS), potassium phosphate, uridine diphosphate glucuronic acid (UDPGA, >98%), and vildagliptin from Sigma-Aldrich (St. Louis, MO, USA); and rat liver microsomes (RLM, R1000) from XenoTech (Lenexa, KS, USA). All solvents were LC-MS grade from Fisher Scientific (Pittsburgh, PA, USA).

### 2.2. Stability Assay of DPP-4 Inhibitors after Incubation with Cysteine

DPP-4 inhibitors (gemigliptin, linagliptin, saxagliptin, trelagliptin, and vildagliptin, Figure 1) were dissolved in methanol. Each DPP-4 inhibitor (100 μM) was incubated in the presence or absence of L-cysteine (100 mM) in potassium phosphate buffer (100 mM, pH of 7.4) at 37 °C for 120 min. An LC-triple quadrupole MS (LC-MS/MS, LC-MS-8060 system, Shimadzu Corporation, Kyoto, Japan) was used to analyze aliquots of the incubation mixture. A Kinetex C18 column (2.1 mm ID × 100 mm, 2.6 μm, Phenomenex) was used to separate analytes, which were eluted with 0.2 mL/min of mobile phases (A: 20 mM ammonium acetate in water, B: acetonitrile) with a linear gradient of 5% B at 0 min, 10% B at 2.0 min, 90% B at 7.0 min, and 5% B at 7.1 min. The LC-MS/MS system was operated in positive ion mode (ionization voltage: 4.0 kV). The selected reaction monitoring transitions of gemigliptin, linagliptin, saxagliptin, trelagliptin, and vildagliptin were *m/z* 490 > 472, 473 > 420, 316 > 180, 358 > 341, and 304 > 154, respectively. The retention times of gemigliptin, linagliptin, saxagliptin, trelagliptin, and vildagliptin were 6.71, 6.06, 5.42, 5.57, and 5.12 min, respectively. The peak area of each DPP-4 inhibitor was expressed as a percentage in comparison with the control in the absence of L-cysteine. All incubations were conducted in triplicate, and the mean values were used for analysis.

The incubation samples with cysteine were analyzed using the Q Exactive Focus Orbitrap MS system (Thermo Fisher Scientific, Waltham, MA, USA) to identify the cysteine adducts of saxagliptin and vildagliptin. The column and LC conditions for the separation of saxagliptin and its metabolites were the same as those mentioned above. The MS system was operated in positive ion mode. The following parameters were optimized for mass detection: capillary temperature: 320 °C; spray voltage: 3.5 kV; and S-lens RF level: 50.0 V. Data were obtained in full scan (*m/z* 300–600; resolution: 70,000; AGC target: 1 × 10^6^) and parallel reaction monitoring (PRM) (resolution: 35,000; normalized collision energy: 22–45 eV; AGC target: 5 × 10^4^) modes. Xcalibur 4.1 software (Thermo Fisher Scientific, Waltham, MA, USA) was used for all data analyses.

### 2.3. Metabolite Profiling in Rat Liver Microsomes

Liver microsomal incubation mixtures were prepared with 100 μM of saxagliptin and 1 mg/mL of microsomal proteins (RLM) in 100 mM of phosphate buffer (pH of 7.4). L-cysteine (100 mM), glutathione (2.5 mM), or UDPGA (5 mM) was added after pre-incubation with or without 100 mM L-cysteine at 37 °C for 5 min to an NADPH-generating system (1.3 mM of NADP^+^, 3.3 mM of G6P, 1 unit/mL of G6PDH, and 3.3 mM of MgCl_2_) to initiate the microsomal reaction [16]. The reaction was stopped by adding cold acetonitrile after microsomal incubation for 2 h. The supernatant was injected into the Q Exactive Focus Orbitrap MS system (Thermo Fisher Scientific, Waltham, MA, USA) after centrifugation (Eppendorf, Hamburg, Germany).

### 2.4. Cysteine and Glutathione Adduct Identification of 5-Hydroxysaxagliptin

In the presence of L-cysteine (100 mM) or glutathione (100 mM) in phosphate buffer (100 mM, pH of 7.4), 5-Hydroxysaxagliptin (100 μM) was incubated at 37 °C for 2 h [4]. The supernatants were analyzed using high-resolution Orbitrap MS after incubation reaction termination and centrifugation. A Kinetex C18 column was used to separate glutathione adducts, which were eluted with 0.2 mL/min of mobile phases (A: 20 mM ammonium acetate in water, B: acetonitrile) with a linear gradient of 5% B at 0 min, 10% B at 2.0 min, 90% B at 7.0 min, and 5% B at 7.1 min.

### 2.5. Metabolite Profiling in Rats

Male Sprague–Dawley (SD) rats (7 weeks old, 200–225 g) were purchased from SamTako Co. (Osan, Republic of Korea). The animals were acclimatized for one week in an animal facility at the College of Pharmacy, Kyungpook National University (Daegu, Republic of Korea). The Kyungpook National University Animal Care and Use Committee (KNU 2021-0120, 12 July 2021) approved the study protocol, conducted under the National Institutes of Health guidance for the care and use of laboratory animals. The rats were randomized into two groups: a control group and a saxagliptin administration group (100 mg/kg). Saxagliptin was dissolved in saline at 100 mg/kg/2 mL concentrations, and saline of 2 mL/kg was used as the vehicle treatment. The bile duct was cannulated with polyethylene tubes (PE-10, Jungdo, Seoul, Republic of Korea) under isoflurane anesthesia (isoflurane vaporizer to 2% with oxygen flow at 0.8 L/min). Blank bile was collected from each rat for 0.5 h before saxagliptin administration, and then the rats received the vehicle and 100 mg/kg of saxagliptin via intraperitoneal injection (*n* = 3 for each group). Bile samples were then collected for 4 h. Blood sampling (approximately 1 mL) was performed via the abdominal artery using a heparin-treated 3-mL syringe under isoflurane anesthesia at 4 h after saxagliptin administration. The liver tissues were immediately excised, gently washed with ice-cold saline-soaked wet tissue to remove the contaminated blood, and homogenized with 4 volumes of saline. Aliquots of plasma, bile, and 20% liver homogenates were stored at −80 °C until the analysis.

The liver homogenates were vortexed with the same volume of methanol/acetonitrile mixtures (1/1, *v*/*v*) and centrifuged (13,500× *g*, 10 min). The remaining pellets were reextracted twice, and pooled supernatants were dried and reconstituted with 50% methanol in water (100 μL) for LC-MS/MS analysis. Protein concentrations in the pellets were determined using a Bradford protein assay kit (Bio-Rad Laboratories, Hercules, CA, USA) with bovine serum albumin as a standard [17]. The rat bile and plasma (200 μL) samples were vortexed with the same methanol/acetonitrile mixture volume (1/1, *v*/*v*). The mixture was centrifuged after vortexing, and the supernatants were dried to complete dryness and reconstituted with 50% methanol in water (100 μL) for LC-MS/MS analysis. The contents of biliverdin in the bile samples were analyzed by measuring ultraviolet absorbance at 380 nm using a microplate reader (Infinite M200 PRO, TECAN, Mannedorf, Switzerland) after 10-fold dilution with 50% methanol [18].

## 3. Results and Discussions

### 3.1. Stability of DPP-4 Inhibitors under Nonenzymatic Conditions

The irreversible covalent binding of a drug or its metabolites with endogenous proteins seems to initiate immune-mediated DILI [3]. Nucleophilic sites of proteins are the target of electrophiles (the drug itself or its metabolites), and nucleophiles, such as L-cysteine or glutathione, could trap these electrophiles [19]. Therefore, cysteine or glutathione conjugate formation with drugs is considered a marker of covalent binding and possible DILI [7]. In particular, vildagliptin, for which liver injury as an adverse drug reaction has been reported, forms a covalent bond with L-cysteine in an enzyme-independent manner [4].

Four DPP IV inhibitors (saxagliptin, trelagliptin, linagliptin, and gemigliptin) were incubated with L-cysteine in the absence of any enzyme source to identify the possibility of drug–protein conjugate formation. This study used vildagliptin, known to form cysteine conjugate [4], as a positive control. Vildagliptin disappeared after 2 h of incubation in phosphate buffer (Figure 2). This result is comparable with those of previous reports [3,4]. Further, most of the saxagliptin disappeared after incubation, whereas linagliptin and gemigliptin demonstrated no difference (Figure 2). The fact that _L_-cysteine decreased the stability of saxagliptin in the absence of an enzyme source indicates that saxagliptin binds to cysteine in a nonenzymatic manner. Mizuno et al. revealed that the nitrile cyanopyrrolidine moiety in the vildagliptin structure is an essential functional group for the interaction with L-cysteine [4]. Our results confirmed their findings because saxagliptin has a nitrile cyanopyrrolidine moiety (Figure 1). In contrast, linagliptin and gemigliptin, which do not have a nitrile moiety, were stable under incubation with L-cysteine. The reactivity of trelagliptin with L-cysteine was lower than that of vildagliptin and saxagliptin (Figure 1). The poor reactivity of trelagliptin may be due to the poor electrophilic properties of the benzonitrile group [20,21].

Next, we attempted to elucidate the structure of the cysteine conjugate of saxagliptin using high-resolution MS after saxagliptin was incubated with L-cysteine in a phosphate buffer. Mizuno et al. reported [4] that the cysteine conjugate of vildagliptin was observed at *m/z* 408.1952 (4.25 min), 105 Dalton (Da) higher than that of vildagliptin (Figure 3a). Fragmentation of the protonated molecular ion [M+H]^+^ showed characteristic fragment ions at *m/z* 346.1940, 258.0899, 241.0634, and 151.1112, which were observed in a previous study (Figure 3a) [4]. Peaks corresponding to the cysteine conjugate of saxagliptin were detected at a retention time of 4.40 min in positive ionization mode after incubation of saxagliptin with L-cysteine for 120 min (Figure 3b). The cysteine conjugate of saxagliptin had a protonated molecular ion at *m/z* 420.1939 (mass error < 3.8 ppm), 105 Da higher than that of saxagliptin. The MS/MS spectrum of the cysteine conjugate fragmented at *m/z* 420.1952 through collision gave the base peak at *m/z* 180.1379, which originated from the cleavage of the amide bond (Figure 3b). Further, the peak at *m/z* 180.0979 was observed in the MS/MS spectrum of saxagliptin and 5-hydroxysaxagliptin [8]. Further, they also gave characteristic fragment ions at *m/z* 402.1833 (loss of a water molecule), 240.0796 (loss of hydroxyadamantane group and CO_2_), and 82.0656 (azabicyclohexane group) (Figure 3b), indicating that the cysteine conjugate of saxagliptin has a thiazoline acid moiety. The fragment ion at *m/z* 132.0112 was expected to be a protonated thiazoline acid (MW = 131.0041). This result reveals that the covalent binding of saxagliptin is an irreversible reaction that forms a thiazoline ring before deamination (Figure 4). This type of cysteine conjugate was also reported for vildagliptin and anagliptin [4].

### 3.2. Metabolite Profiling in Rat Liver Microsomes

This study investigated the metabolism of saxagliptin using RLM. Four metabolites (saxagliptin cyclic amidine [M1], 5-hydroxysaxagliptin [M2], saxagliptin–cysteine conjugate [M3], and 5-hydroxysaxagliptin–cysteine conjugate [M4]) were profiled in PRM mode after the incubation of saxagliptin with RLM in the presence or absence of an NADPH-generating system and/or L-cysteine (Figure 5). Tandem MS analysis of saxagliptin and its four metabolites using high-resolution MS produced prominent and informative product ions for structural elucidation (Figure 6). The protonated molecular ion of saxagliptin was observed at *m/z* 316.2017 (mass error < 1.0 ppm) and eluted at 5.29 min (Figure 5b). The base peak at *m/z* 180.1384 was produced from the cleavage of the amide bond (Figure 6a). M1, M2, M3, and M4 with protonated molecular ions at *m/z* 316.2017 (mass error < 1.0 ppm), 332.1965 (mass error < 1.5 ppm), 420.1931 (mass error < 5.0 ppm), and 436.1895 (mass error < 1.5 ppm) were found at 4.9, 4.4, 4.5, and 3.5 min, respectively (Figure 5).

Metabolite M1 was found in the incubation sample of saxagliptin with buffer solution (Figure 5a) and RLM (Figure 5c) in the presence of +L-cysteine. The major fragment ions at *m/z* 288.2069 and 180.1382 found in M1 (Figure 6b) were designated as characteristic fragment ions of the cyclic amidine degradants of saxagliptin [8]. The protonated molecular ion of metabolite M2 was found at *m/z* 332.1965, indicating that one oxygen atom had been introduced into saxagliptin. Metabolite M2 was observed in rat liver microsomal incubation samples regardless of the addition of L-cysteine; however, the peak intensity of M2 in the absence of -L-cysteine was much higher than that in the presence of +L-cysteine (Figure 5b,c). The characteristic fragment ion of M2 was observed at *m/z* 196.1326, indicating hydroxylation in the adamantane group [8,15] (Figure 6b). Compared with the retention time and fragment ion pattern of the authentic reference compound, M2 was determined as 5-hydroxysaxagliptin (Appendix A). Metabolite M3 was found in the incubation sample of saxagliptin with buffer solution (Figure 5a) and RLM (Figure 5c) in the presence of +L-cysteine, indicating that M3 was formed in an enzyme-nonspecific manner. The MS/MS spectrum of M3 (Figure 6d) was similar to that of the saxagliptin–cysteine conjugate identified under nonenzymatic conditions (Figure 2b) [4]. The protonated molecular ion of metabolite M4 was observed at *m/z* 436.1895, indicating that one oxygen atom might be introduced into the saxagliptin–cysteine conjugate (*m/z* 420.1931). We performed a microsomal incubation study with 5-hydroxysaxagliptin in the presence of L-cysteine to confirm this. Further, M4 was detected in this sample, and both its retention time and fragment ion pattern were nearly identical (Figure 7). The characteristic fragment ions of M4 were found at *m/z* 196.1326 and 213.0686, indicating that hydroxylation occurred in the adamantane group but not in the azabicyclohexane group (Figure 7) [8,15]. Therefore, M1, M2, M3, and M4 were determined as cyclic amidine degradants, 5-hydroxysaxagliptin, saxagliptin–cysteine conjugates, and 5-hydroxysaxagliptin–cysteine conjugates, respectively.

### 3.3. Metabolite Profiling in Rat Plasma, Bile, and Liver

Producing cysteine conjugates of drugs may be a marker of the covalent binding of these drugs to thiol residues of cysteine in endogenous proteins or glutathione [3,7,21]. We revealed that L-cysteine enzyme-independently trapped saxagliptin rapidly to produce the thiazoline-containing cysteine adducts M3 and M4 through an in vitro study. However, no evidence revealed that such thiazoline acid adducts are biotransformed in vivo [8,15]. We attempted to determine these metabolites in plasma, bile, and liver samples after treating rats with saxagliptin.

Seven metabolites were found in rat bile, plasma, and liver tissues after intraperitoneal injection of saxagliptin in rats (Figure 8). Tandem MS analysis using a high-resolution MS produced an informative MS/MS spectrum for the structural identification of metabolites. The MS/MS spectra and retention times of P (*m/z* 316.2020), M1 (*m/z* 316.2020), M2 (*m/z* 332.1969), and M3 (*m/z* 420.1952) were similar to those of saxagliptin, saxagliptin cyclic amidine, 5-hydroxysaxagliptin, and saxagliptin–cysteine conjugate identified in the microsomal metabolism study, respectively (Appendix A). Further, saxagliptin cyclic amidine (M1) and 5-hydroxysaxagliptin (M2) have been determined in human and rat plasma samples [8,15].

Metabolites M5, M6, M7, and M8 with protonated molecular ions at *m/z* 477.2166, 493.2115, 396.1588, and 492.2340 were observed at 4.9, 3.9, 4.6, and 4.3 min, respectively (Figure 8d–g). Metabolite M5 was found in rat bile, plasma, and liver tissues. The base peak at *m/z* 358.2114 was generated from the loss of amine and acetyl groups and the loss of the CH_2_S group in the thiazoline ring. The structures of the other fragment ions at *m/z* 316.2010, 222.1490, and 180.1378 are presented in Appendix A. Metabolite M5 was determined as a cysteinylglycine (Cys–Gly) conjugate of saxagliptin based on the interpreted fragment ion structure. The protonated molecular ion of M6 was observed at *m/z* 493.2115, indicating that one oxygen atom had been introduced into the saxagliptin–Cys–Gly conjugate. Fragment ions observed at *m/z* 374.2068, 332.1960, 238.1432, and 196.1327 revealed an increase of 16 Da compared with fragment ions found in M5 (Appendix A). The characteristic fragment ion at *m/z* 196.1327 indicated that hydroxylation occurred in the adamantine group. Based on these findings, M6 was designated as a 5-hydroxysaxagliptin–Cys–Gly conjugate. Cysteinylglycine conjugate was revealed in vildagliptin-fed mice [3].

Metabolite M7 was only found in rat bile, whereas M8 was observed in rat bile, plasma, and liver tissues (Figure 8). The presence of MH^+^ ions at *m/z* 396.1588 (M7, addition of 80 Da to parent) (mass error < 1.0 ppm) and *m/z* 492.2340 (M8, addition of 176 Da to parent) (mass error < 2.0 ppm) was found through full scan analysis of biological samples obtained after the intraperitoneal injection of saxagliptin in rats, indicating sulfate and glucuronide conjugation in rats [8,22,23]. Subsequent collision-induced *m/z* 396 fragmentation generated product ions at *m/z* 260.0947 and 180.1380 derived from the cleavage of the amide bond and further loss of the sulfate group, respectively (Appendix A). The characteristic fragment ion at *m/z* 260.0947 was found in the MS/MS spectrum of saxagliptin-*O*-sulfate, which was observed in humans [8]. Subsequent collision-induced *m/z* 492 fragmentation generated product ions at *m/z* 316.2013, caused by the typical neutral loss of glucuronide (176 Da) [8,24]. The fragment ions at *m/z* 298.1908 and 281.1643 were derived from the loss of water and further deamination, respectively. Further, characteristic fragment ions at *m/z* 316.2013 and 298.1908 were reported in the MS/MS spectrum of saxagliptin-*O*-glucuronide, which was observed in humans [8]. Fragment ions at *m/z* 180.1379 and 162.1274 in the MS/MS spectrum were derived from the cleavage of the amide bond from the fragment ion at *m/z* 316.2013 and further loss of the water molecule, respectively (Appendix A)**.** Therefore, M7 and M8 were determined as saxagliptin-*O*-sulfate and saxagliptin-*O*-glucuronide, respectively.

Table 1 summarizes the retention times, theoretical and measured masses, mass errors, and reaction types of saxagliptin and its metabolites. The mass errors were within <5 ppm. Four metabolites (M1–M4) were observed in rat liver microsomal incubation samples, and seven metabolites (M1–M3 and M5–M8) were found in rat bile, plasma, and liver tissues. The metabolic pathways determined included cyclization, hydroxylation, and conjugations with cysteine, glutathione, sulfate, and glucuronide (Figure 9).

### 3.4. Identification of the Reactive Metabolites

The production of cysteine or glutathione conjugates of drugs is a marker of the covalent binding of drugs to thiol residues of cysteine in endogenous proteins [3]. We revealed, in vitro, that L-cysteine rapidly trapped saxagliptin in a nonenzymatic manner to produce the thiazoline-containing cysteine adduct (M3). Next, we attempted to clarify whether these metabolites were biotransformed in vivo. After treating rats with saxagliptin, in addition to M3, the cysteinylglycine (Cys–Gly) conjugate of saxagliptin and 5-hydroxysaxagliptin (M5 and M6, respectively) were observed in rats (Appendix A). Mizuno et al. (2019) revealed that the thiazoline moiety of the cysteinylglycine conjugate of vildagliptin was formed by the nucleophilic conjugation of a thiol residue of glutathione to the nitrile moiety before hydrolysis of the cysteine–glutamine bond of glutathione and loss of the ammonia group [3]. Therefore, the thiazoline moiety in M5 and M6 may be formed by the nucleophilic conjugation of a thiol residue of glutathione to the nitrile moiety, similar to vildagliptin (Figure 10). The present study revealed four thiol conjugates (M3–M6) in vitro and in vivo, and these metabolites were generated via cysteine or glutathione conjugation to saxagliptin.

Some drugs covalently bind to the L-cysteine of endogenous proteins in humans. In particular, reactive metabolites of acetaminophen [25] and raloxifene [26] covalently bind to the cysteine residue of endogenous proteins in humans [3]. Saxagliptin and its metabolites irreversibly bind to L-cysteine or glutathione; thus, they may covalently bind to cysteine residues of endogenous proteins in animals and humans, thereby triggering adverse immune reactions. Immune-mediated hepatotoxicity has been reported for vildagliptin, which has a cyanopyrrolidine group like saxagliptin [9], in clinics. The levels of serum transaminase were increased in patients taking vildagliptin (50 mg/day) in Japan, and the drug-induced lymphocyte stimulation test was positive [9]. Dahal et al. [27] and Mizuno et al. [4] reported that radiolabeled vildagliptin binds to macromolecules in human hepatocytes and vildagliptin binds covalently to the thiol residue of L-cysteine in humans, respectively, suggesting that vildagliptin can form covalent bonds with endogenous proteins. Collectively, analogous covalent binding to protein thiols may initiate immune-mediated hepatotoxicity. As for saxagliptin, one case of hepatotoxicity has been reported to date. Thalha et al. (2018) revealed that Kombiglyze (metformin with saxagliptin) induced cholestasis in a patient with nonalcoholic steatohepatitis [13]. Therefore, further studies are required to confirm the covalent binding of saxagliptin to the cysteine residues of endogenous proteins in animals and humans to precisely predict the potential of DILI from saxagliptin in clinics.

### 3.5. Limitations of the Study

The clinical data are very important for the prediction of the DILI potential of saxagliptin. However, we only identified reactive metabolites in rat liver microsomes and rats. Therefore, further studies for the confirmation of reactive metabolites through the incubation study with saxagliptin and human liver microsomes or human hepatocytes are needed. In addition, we did not identify direct binding of the saxagliptin to endogenous proteins although we identified cysteine or glutathione conjugates of saxagliptin in rats. Further studies are also required on the covalent binding of saxagliptin to the cysteine residues of endogenous proteins in animals.

## 4. Conclusions

This study investigated cysteine and glutathione conjugate formation of saxagliptin in rat liver microsomes to elucidate the potential of saxagliptin-induced liver injury. Saxagliptin, a cyanopyrrolidine-containing DPP-4 inhibitor, nonenzymatically binds to L-cysteine to generate a thiazoline-containing cysteine adduct similar to vildagliptin, whereas linagliptin, gemigliptin, and trelagliptin, which contain no cyanopyrrolidine ring, are stable under incubation with L-cysteine in phosphate buffer. Thiazoline-containing cysteine (M3 and M4) and glutathione conjugates (M5 and M6) were observed in rat liver microsomes, bile, plasma, and liver tissues. Our results reveal that saxagliptin covalently binds to the thiol groups of cysteine residues of endogenous proteins in rats, indicating the potential for saxagliptin to cause drug-induced liver injury.

## Figures and Tables

**Figure 1 pharmaceutics-16-00106-f001:**
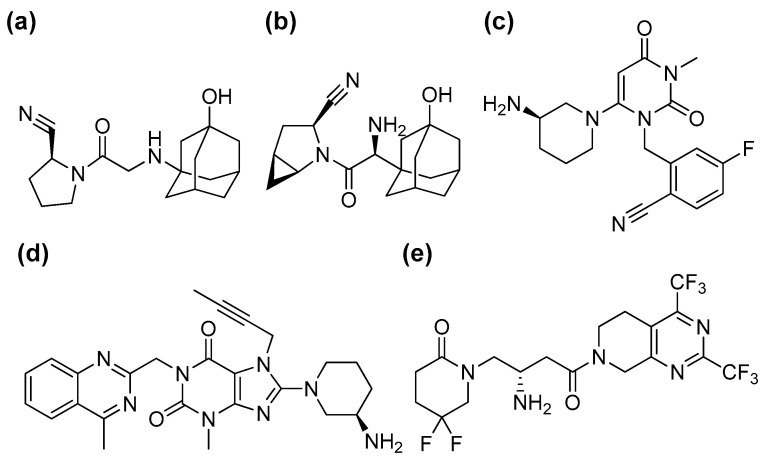
Chemical structures of five dipeptidyl peptidase-4 inhibitors: vildagliptin (**a**), saxagliptin (**b**), trelagliptin (**c**), linagliptin (**d**), and gemigliptin (**e**).

**Figure 2 pharmaceutics-16-00106-f002:**
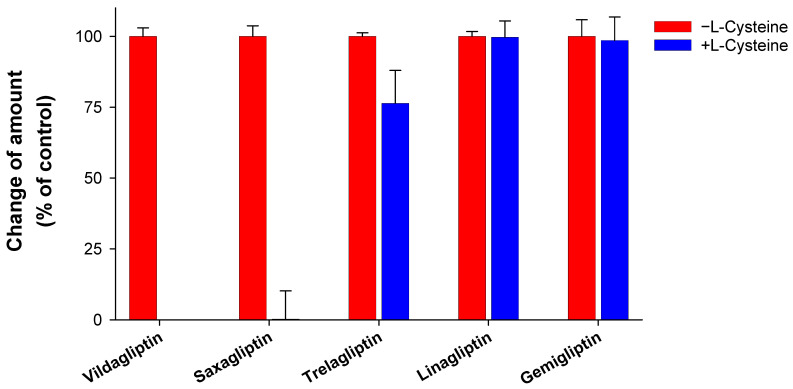
Stability of dipeptidyl peptidase-4 (DPP-4) inhibitors in the presence of L-cysteine. DPP-4 inhibitors were incubated in the presence of L-cysteine for 2 h at 37 °C in potassium phosphate buffer (pH of 7.4). The group without L-cysteine was used as a control. Data are shown as the mean ± standard deviation (SD) of triplicate determinations.

**Figure 3 pharmaceutics-16-00106-f003:**
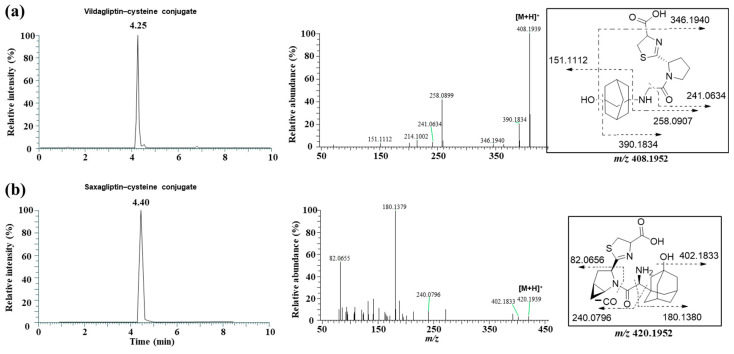
LC-MS/MS analysis to identify the cysteine conjugate of vildagliptin (**a**) and saxagliptin (**b**). Single-ion monitoring chromatogram (left panel) and MS/MS spectrum (right panel) of the cysteine conjugate of vildagliptin (*m/z* 408.1952) and saxagliptin (*m/z* 420.1952).

**Figure 4 pharmaceutics-16-00106-f004:**
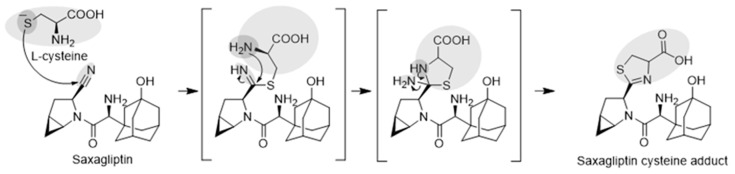
Expected mechanisms for the irreversible covalent binding of L-cysteine to saxagliptin.

**Figure 5 pharmaceutics-16-00106-f005:**
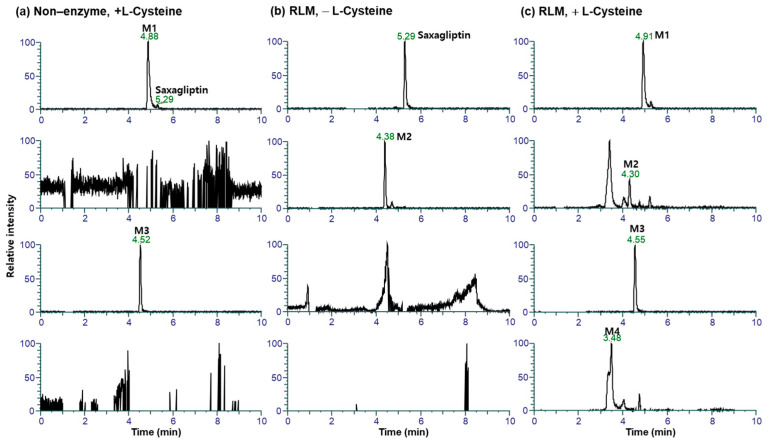
Representative extracted ion chromatograms of saxagliptin and its metabolites generated from liquid chromatography–high-resolution mass spectrometry analysis of rat liver microsomal incubates with saxagliptin and an NADPH-generating system in the presence (**a**) and absence (**b**) of L-cysteine, and incubation samples with saxagliptin in phosphate buffer (pH of 7.4) containing L-cysteine (**c**).

**Figure 6 pharmaceutics-16-00106-f006:**
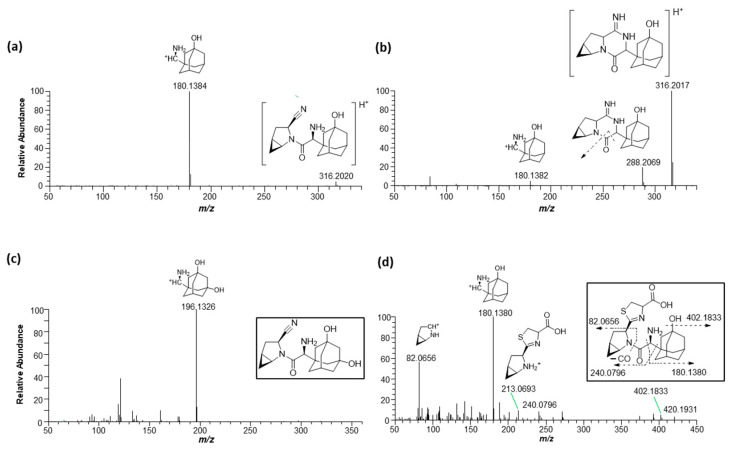
Representative product ion scan mass spectra of saxagliptin (**a**), cyclic amidine degradant (**b**), 5-hydroxysaxagliptin (**c**), and saxagliptin–cysteine conjugate (**d**) obtained from liquid chromatography–high-resolution mass spectrometry analysis of rat liver microsomal incubates with saxagliptin in the presence of an NADPH-generating system and L-cysteine.

**Figure 7 pharmaceutics-16-00106-f007:**
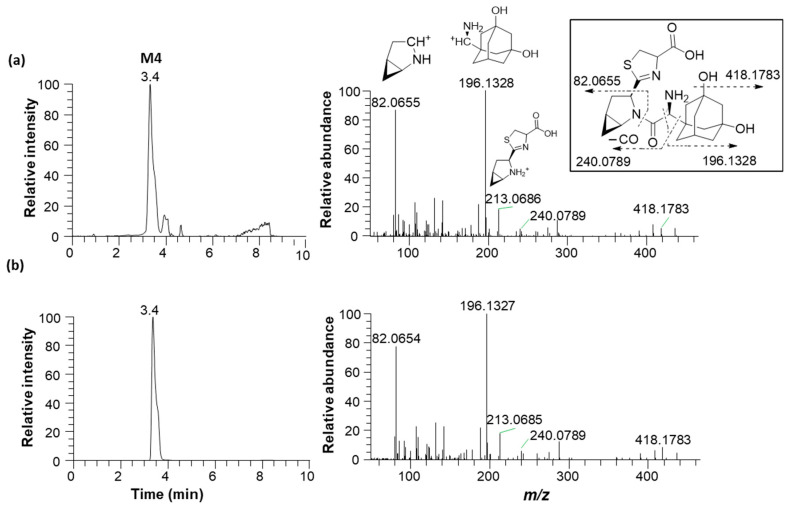
Representative extracted ion chromatograms (*m/z* 436.2020, left panel) and product ion scan mass spectra (right panel) of 5-hydroxysaxagliptin–cysteine conjugate (M4) obtained from liquid chromatography–high-resolution mass spectrometry analysis of rat liver microsomes incubated with saxagliptin (**a**) and 5-hydroxysaxagliptin (**b**) in the presence of L-cysteine.

**Figure 8 pharmaceutics-16-00106-f008:**
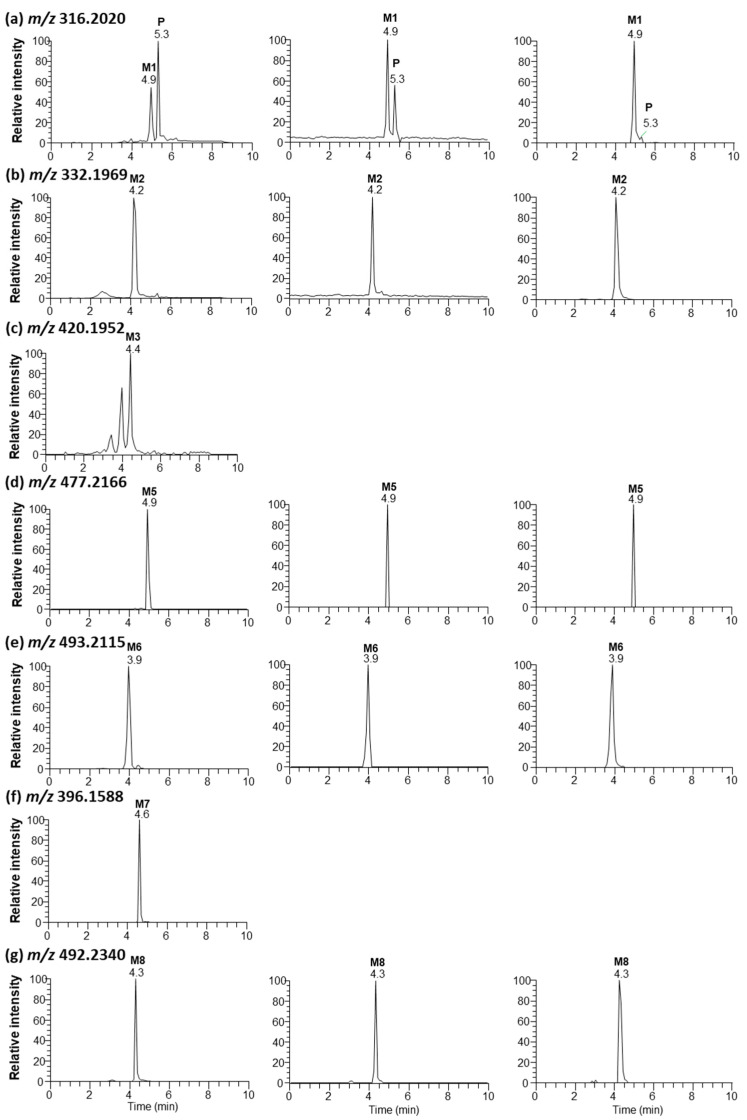
Representative extracted ion chromatograms of saxagliptin (P) and its metabolites (saxagliptin cyclic amidine [M1], 5-hydroxysaxagliptin [M2], saxagliptin–cysteine conjugate [M3], saxagliptin–cysteinylglycine conjugate [M5], 5-hydroxysaxagliptin–cysteinylglycine conjugate [M6], saxagliptin-*O*-sulfate [M7], and saxagliptin-*O*-glucuronide [M8]) obtained from liquid chromatography–high-resolution mass spectrometry analysis of bile (left panel), plasma (center panel), and liver tissues (right panel) obtained from rats after oral administration of saxagliptin (100 mg/kg).

**Figure 9 pharmaceutics-16-00106-f009:**
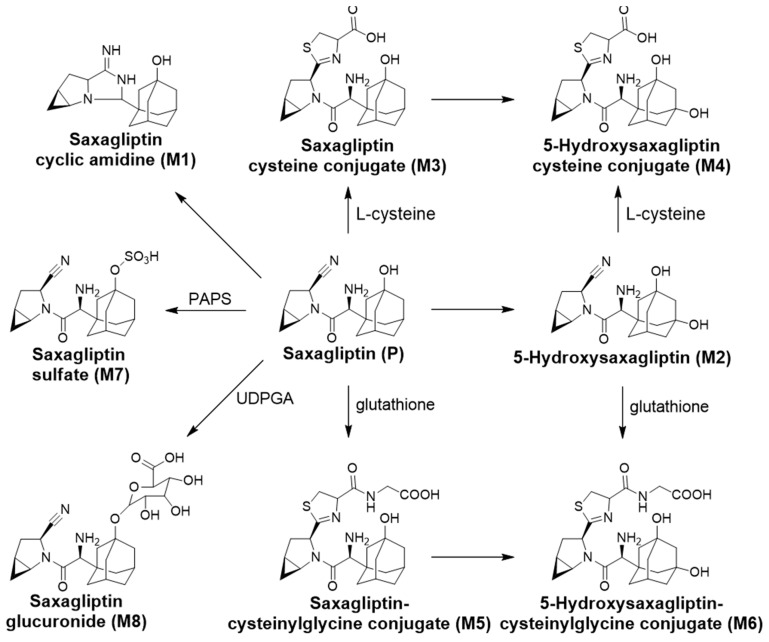
Saxagliptin metabolic pathways proposed in rat liver microsomes and rats.

**Figure 10 pharmaceutics-16-00106-f010:**
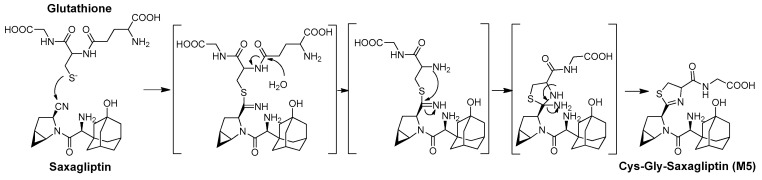
Proposed mechanisms for the irreversible covalent bonding of glutathione to saxagliptin and 5-hydroxysaxagliptin.

**Table 1 pharmaceutics-16-00106-t001:** Summary of the mass spectral data of saxagliptin and its metabolites.

No.	Metabolites	t_R_ (min)	[M+H]^+^		Mass Error (ppm)	Reaction Type
			Theoretical	Measured		
P	Saxagliptin	5.3	316.2020	316.2017	−0.95	
M1	Saxagliptin cyclic amidine	4.9	316.2020	316.2017	−0.95	cyclization
M2	5-Hydroxysaxagliptin	4.2	332.1969	332.1965	−1.2	hydroxylation
M3	Saxagliptin–cysteine conjugate	4.4	420.1952	420.1931	−4.99	cysteine conjugation
M4	5-Hydroxysaxagliptin–cysteine conjugate	3.5	436.1901	436.1895	−1.40	Hydroxylation and cysteine conjugation
M5	Saxagliptin–cysteinylglycine conjugate	4.7	477.2166	477.2163	−0.63	glutathione conjugation
M6	5-Hydroxysaxagliptin–cysteinylglycine conjugate	3.9	493.2115	493.2113	−0.41	Hydroxylation and glutathione conjugation
M7	Saxagliptin-*O*-sulfate	4.6	396.1588	396.1581	−1.77	sulfation
M8	Saxagliptin-*O*-glucuronide	4.3	492.2340	492.2335	−1.01	glucuronidation

## Data Availability

All data in this study have been included in this manuscript.

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
