# Peer review of "Evaluation of the Drug-Induced Liver Injury Potential of Saxagliptin through Reactive Metabolite Identification in Rats"

_pharmaceutics, 2024, doi:10.3390/pharmaceutics16010106_

Round 1

Reviewer 1 Report

Comments and Suggestions for Authors

The study is very informative and interesting. Table 1 is  a very informative and precise. The potential of saxagliptin-induced liver injury is well proposed and illustrated. However, many comments should be addressed.

1-     The following articles should be cited in the introduction

Humphreys WG. Investigating the link between drug metabolism and toxicity. InOvercoming Obstacles in Drug Discovery and Development 2023 Jan 1 (pp. 201-213). Academic Press.

Halegoua-De Marzio D, Navarro VJ. Hepatotoxicity of cardiovascular and antidiabetic drugs. InDrug-induced liver disease 2013 Jan 1 (pp. 519-540). Academic Press.

Helal MG, Zaki MM, Said E. Nephroprotective effect of saxagliptin against gentamicin-induced nephrotoxicity, emphasis on anti-oxidant, anti-inflammatory and anti-apoptic effects. Life sciences. 2018 Sep 1;208:64-71.

2-     The size and resolution of fig 1 should be improved to be more visible for readers

3-     Correct numbering for sections, 1.2 , 1.3 ,…. To be 2.2, 2.3,……………

4-     In section 1.2 add reference for LC-MS/MS conditions.

5-     Also , add reference for A Kinetex C18 column was used to separate saxagliptin and its metabo-lites

6-     Add reference for 1.3. Metabolite Profiling in Rat Liver Microsomes. The same for 1.4 and 1.5

7-     In section 1.5, number of rats in each group should be stated

8-     Figure 2 should be displayed with colors for more visibility

9-     Full name for 105 Da should be stated in page. also for RLM, PRM

10-  In figure 5 description add + for L-cysteine to be +L-cysteine

11-  Figure S1 should be referred in the text

12-  Recommendations about safety of ant diabetic drugs according to the presented study findings and comparisons with literature reviews should be provided in a separate section

13-  Study limitation and future research plans should be provided

remove 5- patents in line 189

Best wishes

Author Response

  1. The following articles should be cited in the introduction

- Humphreys WG. Investigating the link between drug metabolism and toxicity. InOvercoming Obstacles in Drug Discovery and Development 2023 Jan 1 (pp. 201-213). Academic Press.

- Halegoua-De Marzio D, Navarro VJ. Hepatotoxicity of cardiovascular and antidiabetic drugs. InDrug-induced liver disease 2013 Jan 1 (pp. 519-540). Academic Press.

- Helal MG, Zaki MM, Said E. Nephroprotective effect of saxagliptin against gentamicin-induced nephrotoxicity, emphasis on anti-oxidant, anti-inflammatory and anti-apoptic effects. Life sciences. 2018 Sep 1;208:64-71.

→ We appreciate your kind comment. As you commented, we added this references in our revised manuscript.

  1. The size and resolution of fig 1 should be improved to be more visible for readers

→ Thank you for your comment. As you commented, we modified the size and resolution of Figure 1 in our revised manuscript.

  1. Correct numbering for sections, 1.2 , 1.3 ,…. To be 2.2, 2.3,……………

→ Thank you for your comment. As you commented, we corrected Section numbering in our revised manuscript.

  1. In section 1.2 add reference for LC-MS/MS conditions.

→ We used the MS/MS conditions for the analysis of saxagliptin and its metabolites developed by ourselves.

  1. Also, add reference for A Kinetex C18 column was used to separate saxagliptin and its metabo-lites

→ We used the chromatographic conditions for the analysis of saxagliptin and its metabolites developed by ourselves.

  1. Add reference for 1.3. Metabolite Profiling in Rat Liver Microsomes. The same for 1.4 and 1.5

→ Thank you for your comment. As you commented, we added reference information in our revised manuscript.

  1. In section 1.5, number of rats in each group should be stated

→ We already stated in section 2.5 as follows:

Blank bile was collected from each rat for 0.5 h before saxagliptin admin-istration, and then the rats received vehicle and 10, 50, and 100 mg/kg of saxagliptin via intraperitoneal injection (n = 3 for each group).

  1. Figure 2 should be displayed with colors for more visibility

→ As you commented, we modified Figure 2 in our revised manuscript.

  1. Full name for 105 Da should be stated in page. also for RLM, PRM

→ We appreciate your kind comment. As you commented, we added this information in our revised manuscript. Full name of RLM and PRM already stated in Section 2.1 and 2.2 of our original manuscript.

Mizuno et al. reported [4] that the cysteine conjugate of vildagliptin was observed at m/z 408.1952 (4.25 min), 105 Da higher than that of vildagliptin (Figure 3a).

→ Mizuno et al. reported [4] that the cysteine conjugate of vildagliptin was observed at m/z 408.1952 (4.25 min), 105 Dalton (Da) higher than that of vildagliptin (Figure 3a).

  1. In figure 5 description add + for L-cysteine to be +L-cysteine

→ Thank you for your comment. As you commented, we added this information in our revised manuscript.

Metabolite M1 was found in the incubation sample of saxagliptin with buffer solution (Figure 5a) and RLM (Figure 5c) in the presence of L-cysteine.

→ Metabolite M1 was found in the incubation sample of saxagliptin with buffer solution (Figure 5a) and RLM (Figure 5c) in the presence of +L-cysteine.

however, the peak intensity of M2 in the absence of L-cysteine was much higher than that in the presence of L-cysteine (Figures 5b and 5c).

→ however, the peak intensity of M2 in the absence of -L-cysteine was much higher than that in the presence of +L-cysteine (Figures 5b and 5c).

Metabolite M3 was found in the incubation sample of saxagliptin with buffer solution (Figure 5a) and RLM (Figure 5c) in the presence of L-cysteine, indicating that M3 was formed in an enzyme-nonspecific manner.

→ Metabolite M3 was found in the incubation sample of saxagliptin with buffer solution (Figure 5a) and RLM (Figure 5c) in the presence of +L-cysteine, indicating that M3 was formed in an enzyme-nonspecific manner.

  1. Figure S1 should be referred in the text

→ We already stated in section 3.2 as follows:

Compared with the retention time and fragment ion pattern of the authen-tic reference compound, M2 was determined as 5-hydroxysaxagliptin (Figure S1).

  1. Recommendations about safety of ant diabetic drugs according to the presented study findings and comparisons with literature reviews should be provided in a separate section

→ Thank you for your valuable comment. As you commented, we added this information in final paragraph of “3.4. Identification of the Reactive Metabolites” Part of “Results and Discussion” Section of our revised manuscript as follows.

Some drugs covalently bind to the L-cysteine of endogenous proteins in humans. In particular, reactive metabolites of acetaminophen and raloxifene covalently bind to the cysteine residue of endogenous proteins in humans. Saxagliptin and its metabolites irreversibly bind to L-cysteine or glutathione, thus it may covalently bind to cysteine residues of endogenous proteins in animals and humans, thereby triggering adverse immune reactions. Immune-mediated hepatotoxicity has been reported for vildagliptin which has a cyanopyrrolidine group like saxagliptin in clinics. The levels of serum transaminase were increased in patients taking vildagliptin (50 mg/day) in Japan, and the drug-induced lymphocyte stimulation test was positive. Dahal et al. and Mizuno et al. reported that radiolabeled vildagliptin binds to macromolecules in human hepatocytes and vildagliptin binds covalently to the thiol residue of L-cysteine in humans, respectively, suggesting that vildagliptin can form covalent bonds with endogenous proteins. Collectively, analogous covalent binding to protein thiols may initiate immune-mediated hepatotoxicity. As for saxagliptin, one case of hepatotoxicity has been reported to date. Thalha et al. (2018) revealed that Kombiglyze (metformin with saxagliptin) induced cholestasis in a patient with nonalcoholic steatohepatitis. Therefore, further studies are required to confirm the covalent binding of saxagliptin to the cysteine residues of endogenous proteins in animals and humans to precisely predict the potential of DILI from saxagliptin in clinics.

  1. Study limitation and future research plans should be provided

remove 5- patents in line 189

→ Thank you for your valuable comment. As you commented, we added study limitation and future plans in “3.5 Limitations of the Study” Part of our revised manuscript as follows.

3.5. Limitations of the Study

The clinical data is very important for the prediction of DILI potential of saxagliptin. However, we only identified reactive metabolites in rat liver mi-crosomes and rats. Therefore, further studies for the confirmation of reactive metabolites through the incubation study with saxagliptin and human liver microsomes or human hepatocytes are needed. In addition, we did not identify direct binding of the saxagliptin to endogeneous proteins although we identified cysteine or glutathione conjugates of saxagliptin in rats. Further studies are also required to covalent binding of saxagliptin to the cysteine residues of endogeneous proteins in animals.

In addition, we removed “5- patents” in our revised manuscript.

Reviewer 2 Report

Comments and Suggestions for Authors

Comment

Date: 29-12-2023

Manuscript ID: pharmaceutics-2822130

Kim et al. addressed very interesting findings in their research article entitle as “Evaluation of the drug-induced liver injury potential of saxagliptin through reactive metabolite identification in rats”. The work is interesting, well designed, and informative to reader working in the domains. However, I recommend few suggestions before publication.

Comment 1: In abstract “Vildagliptin-induced liver injury has been reported in rare cases. Vildagliptin binds covalently to L-cysteine or glutathione to generate thia-zoline-containing thiol adducts, indicating that covalent binding may cause immune-mediated hepatotoxicity. A liver injury was recently reported for saxagliptin, which is a dipeptidyl peptidase-4 (DPP-4) inhibitor. However, the underlying mechanisms of saxagliptin-induced liver injury remain unknown. Similar to vildagliptin, saxagliptin demonstrated a reactive cyanopyrrolidine moiety, indicating the possibility of thiazoline-containing thiol metabolite formation in vivo” I suggest to shift this content to introduction section. The abstract is qualitative rather than quantitative explanation. The authors are recommended to restructure the abstract and explain from experimental to the result-discussion as obtained in the study. It seems to be more introductory than the concise information of the article. Please replace nucleophiles with “the nucleophiles” and define in the abstract section. Similarly, in the sentence “four new thiazoline-containing thiol metabolites”, I suggest to define these four in the bracket.

 Comment 2: In the sentence “Idiosyncratic reactions caused most DILI cases”, the authors need to justify the claim by citing the referenced article. In material sections, the authors have reported various reagents used in the study. I suggest to report the purity percentage in the revised version of manuscript. Moreover, please mention the grade of the chemicals used in the investigation. Figure 1 needs to be restructured with bold line and clarity. Please compile them one over other. It looks as poor quality image and unclear.   

Comment 3: In section 2.1, the authors should elaborate the column dimension and mobile phase composition. How did the authors optimize the ratio of the mobile phase for the separation of multiple analytes together? Was it validated before? Please define “linear gradient”? The retention time of 5.57 and 5.42 min are very close. How did the authors differentiate them? What was the reason for not using any internal standards? Why is the repetition of “Kinetex C18 column was used to separate saxagliptin and its metabo-lites, which were eluted with 0.2 mL/min of mobile phases (A: 20 mM ammonium acetate in water, B: acetonitrile) with a linear gradient of 5% B at 0 min, 10% B at 2.0 min, 90% B at 7.0 min, and 5% B at 7.1 min.” in the same section?.  

Comment 4: I recommend to provide the source of the instrument used in the study such as centrifugation and others. Please include city, state and country.  

Comment 5: In section 1.4, I found “A Kinetex C18 column was used to separate glutathione adducts, which were eluted with mobile phases (A: 20 mM ammonium acetate in water, B: acetonitrile”. I suggest to explain percent composition of both components of the mobile phase.      

Comment 6: I found variation in incubation time for different samples. Please explain.

Comment 7: In figure 2, I suggest to keep consistency in font size. From figure 3a and figure 4, remove the text “figure”. In figure 3, I suggest to label m and m+1 peak.

Comment 8: Provide figure 9 and 10 as supplementary figures in the revised file.  

Comment 9: In the sentence “Our results reveal that 185 saxagliptin covalently binds to the thiol groups of cysteine residues of endogenous proteins in animals and humans, indicating the use of saxagliptin for drug-induced liver 187 injury,” I did not find any human data.

Comments on the Quality of English Language

The language is good and acceptable.

Author Response

  1. In abstract “Vildagliptin-induced liver injury has been reported in rare cases. Vildagliptin binds covalently to L-cysteine or glutathione to generate thia-zoline-containing thiol adducts, indicating that covalent binding may cause immune-mediated hepatotoxicity. A liver injury was recently reported for saxagliptin, which is a dipeptidyl peptidase-4 (DPP-4) inhibitor. However, the underlying mechanisms of saxagliptin-induced liver injury remain unknown. Similar to vildagliptin, saxagliptin demonstrated a reactive cyanopyrrolidine moiety, indicating the possibility of thiazoline-containing thiol metabolite formation in vivo” I suggest to shift this content to introduction section. The abstract is qualitative rather than quantitative explanation. The authors are recommended to restructure the abstract and explain from experimental to the result-discussion as obtained in the study. It seems to be more introductory than the concise information of the article. Please replace nucleophiles with “the nucleophiles” and define in the abstract section.

→ Thank you for your comments. We modified “Abstract” based on your comments.

  1. Similarly, in the sentence “four new thiazoline-containing thiol metabolites”, I suggest to define these four in the bracket.

→ Thank you for your comments. We added metabolites information in our revised manuscript as follows.

In summary, we determined four new thiazoline-containing thiol metabolites in saxagliptin-administered male rats.

→ In summary, we determined four new thiazoline-containing thiol metabolites (cysteine and cysteinylglycine conjugates of saxagliptin and 5-hydroxysaxagliptin) in saxagliptin-administered male rats.

  1. In the sentence “Idiosyncratic reactions caused most DILI cases”, the authors need to justify the claim by citing the referenced article.

→ Thank you for your valuable comment. We already stated in “Introduction” section as follows:

The irreversible covalent binding of a drug or its reactive metabolites to endogenous proteins cause idiosyncratic DILI, and drug–protein adducts acquire immunogenicity as antigens and illicit immunological responses [3-5]. Some drugs, such as aspirin and ampicillin, covalently bind to proteins [6]. Reactive intermediates produced by the cytochrome P450 enzyme directly bind to proteins for other drugs, such as acetaminophen [6]. Nucleophilic sites of endogenous proteins, such as the thiol residue of cysteine, are targets for drugs and their electrophilic intermediates. N-acetyl-L-cysteine, L-cysteine, or glutathione trapped these reactive electrophiles. Cysteine and glutathione conjugates of drugs are considered markers of covalent binding with endogenous proteins [4,7].

  1. In material sections, the authors have reported various reagents used in the study. I suggest to report the purity percentage in the revised version of manuscript. Moreover, please mention the grade of the chemicals used in the investigation.

→Thank you for your comment. As you commented, we added these information in our revised manuscript.

  1. Figure 1 needs to be restructured with bold line and clarity. Please compile them one over other. It looks as poor quality image and unclear.

→Thank you for your comment. As you commented, we modified the size and resolution of Figure 1 in our revised manuscript.

  1. In section 2.1, the authors should elaborate the column dimension and mobile phase composition. How did the authors optimize the ratio of the mobile phase for the separation of multiple analytes together? Was it validated before? Please define “linear gradient”?

→ Thank you for your comment. In this study, we didn’t perform quantitative analysis. Therefore, method validation was not needed.

  1. The retention time of 5.57 and 5.42 min are very close. How did the authors differentiate them?

→ The retention times of saxagliptin and trelagliptin were 5.42 and 5.57, respectively, in this study. However, the SRM (selected reaction monitoring) conditions of them are different (316 > 180 for saxagliptin, 358> 341 for trelagliptin). The SRM condition is very specific for analyte, therefore, we can differentiate them.

  1. What was the reason for not using any internal standards?

→ Thank you for your comment. In this study, we performed qualitative analysis. In general, internal standard was not needed for qualitative analysis.

  1. Why is the repetition of “Kinetex C18 column was used to separate saxagliptin and its metabolites, which were eluted with 0.2 mL/min of mobile phases (A: 20 mM ammonium acetate in water, B: acetonitrile) with a linear gradient of 5% B at 0 min, 10% B at 2.0 min, 90% B at 7.0 min, and 5% B at 7.1 min.” in the same section?

→ As you commented, we deleted the repeated condition in our revised manuscript.

  1. I recommend to provide the source of the instrument used in the study such as centrifugation and others. Please include city, state and country.

→ Thank you for your comment. As you commented, we added the source information of the instruments in our revised manuscript (centrifuge, microplate reader and so on).

  1. In section 1.4, I found “A Kinetex C18 column was used to separate glutathione adducts, which were eluted with mobile phases (A: 20 mM ammonium acetate in water, B: acetonitrile”. I suggest to explain percent composition of both components of the mobile phase.

→ As you commented, we added the mobile phase condition in our revised manuscript.

which were eluted with mobile phases (A: 20 mM ammonium acetate in water, B: acetonitrile).

→ which was eluted with 0.2 mL/min of mobile phases (A: 20 mM ammo-nium acetate in water, B: acetonitrile) with a linear gradient of 5% B at 0 min, 10% B at 2.0 min, 90% B at 7.0 min, and 5% B at 7.1 min.

  1. I found variation in incubation time for different samples. Please explain.

→ I am sorry there was typo in original manuscript. Incubation time was 2 h for cysteine or glutathione adduct formation experiment. I corrected this type in our revised manuscript.

  1. In figure 2, I suggest to keep consistency in font size. From figure 3a and figure 4, remove the text “figure”. In figure 3, I suggest to label m and m+1 peak.

→Thank you for your comment. As you commented, we modified figures in our revised manuscript.

  1. Provide figure 9 and 10 as supplementary figures in the revised file.

→ Thank you for your valuable comment. As you commented, we moved Fig. 9 and Fig 10 to supplementary data.

  1. In the sentence “Our results reveal that saxagliptin covalently binds to the thiol groups of cysteine residues of endogenous proteins in animals and humans, indicating the use of saxagliptin for drug-induced liver injury,” I did not find any human data.

→Thank you for your comment. As you commented, we modified this sentence as follows.

Our results reveal that saxagliptin covalently binds to the thiol groups of cysteine residues of endogenous proteins in animals and humans, indicating the use of saxagliptin for drug-induced liver injury.

→ Our results reveal that saxagliptin covalently binds to the thiol groups of cysteine residues of endogenous proteins in rats, indicating the potential for saxagliptin to cause drug-induced liver injury.

Reviewer 3 Report

Comments and Suggestions for Authors

Review of Manuscript ID: pharmaceutics-2822130
Type of manuscript: Article
Title: Evaluation of the drug-induced liver injury potential of saxagliptin
through reactive metabolite identification in rats
Authors: Ki-Young Kim, Yeo-Jin Jeong, So-Young Park, Eun-Ji Park, Ji-Hyeon
Jeon, Im-Sook Song *, And Kwang-Hyeon Liu *
Submitted to section: Pharmacokinetics and Pharmacodynamics

Summary: The paper presents evidence derived from in vitro non-enzymatic and rat liver microsome incubations, as well as from a rat study at different doses of saxagliptin to identify cysteine and glutathione adducts of saxagliptin and its 5-OH metabolite. The authors suggest that based on their findings, saxagliptin has the potential for DILI. The authors reference similar work done with vildagliptin and use this drug as a positive control. They also evaluate other DPP-4 inhibitors that have not reported liver injury associated with their use.

The work is basically an extension of vildagliptin work, implicating the cyanopyrrolidine substructure as the substrate for thiol nucleophilic conjugation. The work appears mostly sound in its design and interpretation. It would have greater impact had human microsomes been included.

Major comments: There is no information regarding the source of results presented in Figures 8, 9 and 10, and Table 1 from rat bile, plasma and liver profiling. There were three doses administered. Which of these doses were used in these figures and their discussion in the text? Was there a dose-responsive increase in the various metabolites observed? This needs to be discussed.

Minor comments:

1.      Abstract: Revise last sentence. Change the word “use” in the latter part of the sentence, “…indicating the use of saxagliptin….”  Suggest changing this to “…indicating the potential for saxagliptin to cause drug-induced liver injury.”

2.      Introduction: Delete last sentence of paragraph beginning, “Saxagliptin is a potent and selective DPP-4 inhibitor….” This sentence is too strong implicating saxagliptin as a causative agent of DILI, given only one referenced study.

3.      Results and Discussion: Paragraph beginning, “Next, we attempted to….” Reference to Figure 3b in the sentence beginning, “Fragmentation of the protonated molecular ion…” appears to incorrectly refer to Figure 3b when discussing vildagliptin results.

4.      Figure 3b gives a retention time of 4.40, but paragraph noted above gives retention time of 4.44 minutes. Please correct either text or figure so that the numbers are consistent.

5.      Results and Discussion: In paragraph noted above, sentence beginning, “The cysteine conjugate of saxagliptin…” incorrectly refers to vildgliptin at the end of the sentence.

6.      Results and Discussion: Figure 5 a, b, and c designations do not match with legend designations.

7.      Results and Discussion: Text is inconsistent when referencing figures. Sometimes these references are in bold print, while in other instances in normal font. Consistent use of bold font is recommended to help readers identify figure references in the text.

8.      Figure 12: glutathione structure is incomplete for saxagliptin (missing terminal COOH).

9.      It is not necessary to show both mechanisms for saxagliptin and the 5-OH metabolite, since they are identical.

10.  Conclusions: As with last sentence of Abstract, recommend changing the structure of the last sentence of this section.

Comments on the Quality of English Language

Minor editing needed for verb tense. 

Author Response

  1. There is no information regarding the source of results presented in Figures 8, 9 and 10, and Table 1 from rat bile, plasma and liver profiling. There were three doses administered. Which of these doses were used in these figures and their discussion in the text? Was there a dose-responsive increase in the various metabolites observed? This needs to be discussed.

→ We appreciate your valuable comments. Figure 8 was from LC-MS/MS analysis of bile, plasma and liver tissues obtained from rats after oral administration of saxagliptin (100 mg/kg). We added this information in our revised manuscript. In this study, we only analyzed samples obtained from 100 mg/kg saxagliptin-administered rats. Therefore, we deleted the information for two groups (10 and 50 mg/kg) in “2.5 Metabolite Profiling in Rats” Part of “Materials and Methods” Section.

  1. Abstract: Revise last sentence. Change the word “use” in the latter part of the sentence, “…indicating the use of saxagliptin….” Suggest changing this to “…indicating the potential for saxagliptin to cause drug-induced liver injury.”

→Thank you for your comment. As you commented, we modified this sentence based on your comment.

  1. Introduction: Delete last sentence of paragraph beginning, “Saxagliptin is a potent and selective DPP-4 inhibitor….” This sentence is too strong implicating saxagliptin as a causative agent of DILI, given only one referenced study.

→ We appreciate your kind comment. As you commented, we deleted this sentence in our revised manuscript.

  1. Results and Discussion: Paragraph beginning, “Next, we attempted to….” Reference to Figure 3b in the sentence beginning, “Fragmentation of the protonated molecular ion…” appears to incorrectly refer to Figure 3b when discussing vildagliptin results.

→ Thank you for your comment. As you commented, we corrected this information in our revised manuscript.

Fragmentation of the protonated molecular ion [M+H]+ showed characteristic fragment ions at m/z 346.1940, 258.0899, 241.0634, and 151.1112, which were observed in a previous study (Figure 3b)

→ Fragmentation of the protonated molecular ion [M+H]+ showed characteristic fragment ions at m/z 346.1940, 258.0899, 241.0634, and 151.1112, which were observed in a previous study (Figure 3a)

  1. Figure 3b gives a retention time of 4.40, but paragraph noted above gives retention time of 4.44 minutes. Please correct either text or figure so that the numbers are consistent.

→ We appreciate your kind comment. As you commented, we corrected this information in our revised manuscript (4.44 min à 4.40 min).

  1. Results and Discussion: In paragraph noted above, sentence beginning, “The cysteine conjugate of saxagliptin…” incorrectly refers to vildgliptin at the end of the sentence.

→ Thank you for your valuable comment. As you commented, we corrected this typo in our revised manuscript (vildagliptin à saxagliptin).

  1. Results and Discussion: Figure 5 a, b, and c designations do not match with legend designations.

→ We appreciate your kind comment. As you commented, we corrected this typo in our revised manuscript (Figure 5a à Figure 5c, Figure 5c à Figure 5a).

  1. Results and Discussion: Text is inconsistent when referencing figures. Sometimes these references are in bold print, while in other instances in normal font. Consistent use of bold font is recommended to help readers identify figure references in the text.

→ Thank you for your comment. As you commented, we unified the font of Figures and Tables as bold font.

  1. Figure 12: glutathione structure is incomplete for saxagliptin (missing terminal COOH).

→ Thank you for your comment. As you commented, we corrected this error in our revised manuscript.

  1. It is not necessary to show both mechanisms for saxagliptin and the 5-OH metabolite, since they are identical.

→ Thank you for your comment. As you commented, we deleted mechanism for 5-hydroxysaxagliptin in our revised manuscript.

  1. Conclusions: As with last sentence of Abstract, recommend changing the structure of the last sentence of this section.

→ Thank you for your valuable comment. As you commented, we modified this sentence in our revised manuscript.

Reviewer 4 Report

Comments and Suggestions for Authors

This paper presents an in vitro and in vivo study of the metabolism in rats of saxagliptin, a dipeptidyl peptidase -4 (DPP-4) inhibitor.  This drug, used for treating diabetes mellitus 2, induces rare cases of saxagliptin-induced liver injury in patients.  The study is largely based on mass spectrometry analysis, but also involved in vitro and in vivo experiments.

Compared to other DPP-4 inhibitors, saxagliptin and vildagliptin, due the presence of a reactive cyanopyrrolidine moiety, conjugate to cysteine in the absence of enzymes. In vitro, in the presence of rat microsomes, cysteine and glutathione conjugates are formed and, in vivo in rats, cysteinyl glycine conjugates of saxagliptin and 5-hydroxysaxagliptin are produced.  The authors conclude “that saxagliptin covalently binds to the thiol groups of cysteine residues of endogenous proteins in animals and humans, indicating the use of saxagliptin for drug-induced liver injury”.

This study is well conducted, well written and well discussed and, therefore, deserves its publication in this journal. 

Nevertheless, I have a few regrets.

·        In their discussion, the authors should come back more completely on the clinical situation associated with the use of saxagliptin and drug-induced liver injury: do they found in the literature indications of a correlation between metabolism of DPP-4 inhibitors, in particular saxagliptin and vildagliptin, formation of adducts and induced disease in patients? Evidence for the presence of antibodies against such conjugates in human serum?

·        For me, the main limitation of this study is that it is restricted to the rat.  It is obvious, at least for ethical reasons, that collecting human bile and making liver biopsies in patients, would not be easy, but having access to serum samples from patients, with or without the liver injury, should remain feasible.  Furthermore, using human microsomes, which are commercially available, and human hepatocytes – primary or continuous cell lines – would have considerably reinforce the scope of this study, at least for the key experiments.  This limitation should be discussed.

·        A minor detail: the clinical application of saxagliptin should be indicated in the abstract to increase the scope of the study.

Author Response

  1. In their discussion, the authors should come back more completely on the clinical situation associated with the use of saxagliptin and drug-induced liver injury: do they found in the literature indications of a correlation between metabolism of DPP-4 inhibitors, in particular saxagliptin and vildagliptin, formation of adducts and induced disease in patients? Evidence for the presence of antibodies against such conjugates in human serum?

→ Thank you for your valuable comment. As you commented, we added this information in final paragraph of “3.4. Identification of the Reactive Metabolites” Part of “Results and Discussion” Section of our revised manuscript as follows.

Some drugs covalently bind to the L-cysteine of endogenous proteins in humans. In particular, reactive metabolites of acetaminophen and raloxifene covalently bind to the cysteine residue of endogenous proteins in humans. Saxagliptin and its metabolites irreversibly bind to L-cysteine or glutathione, thus it may covalently bind to cysteine residues of endogenous proteins in animals and humans, thereby triggering adverse immune reactions. Immune-mediated hepatotoxicity has been reported for vildagliptin which has a cyanopyrrolidine group like saxagliptin in clinics. The levels of serum transaminase were increased in patients taking vildagliptin (50 mg/day) in Japan, and the drug-induced lymphocyte stimulation test was positive. Dahal et al. and Mizuno et al. reported that radiolabeled vildagliptin binds to macromolecules in human hepatocytes and vildagliptin binds covalently to the thiol residue of L-cysteine in humans, respectively, suggesting that vildagliptin can form covalent bonds with endogenous proteins. Collectively, analogous covalent binding to protein thiols may initiate immune-mediated hepatotoxicity. As for saxagliptin, one case of hepatotoxicity has been reported to date. Thalha et al. (2018) revealed that Kombiglyze (metformin with saxagliptin) induced cholestasis in a patient with nonalcoholic steatohepatitis. Therefore, further studies are required to confirm the covalent binding of saxagliptin to the cysteine residues of endogenous proteins in animals and humans to precisely predict the potential of DILI from saxagliptin in clinics.

  1. For me, the main limitation of this study is that it is restricted to the rat. It is obvious, at least for ethical reasons, that collecting human bile and making liver biopsies in patients, would not be easy, but having access to serum samples from patients, with or without the liver injury, should remain feasible. Furthermore, using human microsomes, which are commercially available, and human hepatocytes – primary or continuous cell lines – would have considerably reinforce the scope of this study, at least for the key experiments. This limitation should be discussed.

→ Thank you for your valuable comment. As you commented, we added study limitation and future plans in “3.5 Limitations of the Study” Part of our revised manuscript as follows.

3.5. Limitations of the Study

The clinical data is very important for the prediction of DILI potential of saxagliptin. However, we only identified reactive metabolites in rat liver microsomes and rats. Therefore, further studies for the confirmation of reactive metabolites through the incubation study with saxagliptin and human liver microsomes or human hepatocytes are needed. In addition, we did not identify direct binding of the saxagliptin to endogeneous proteins although we identified cysteine or glutathione conjugates of saxagliptin in rats. Further studies are also required to covalent binding of saxagliptin to the cysteine residues of endogeneous proteins in animals.

  1. A minor detail: the clinical application of saxagliptin should be indicated in the abstract to increase the scope of the study.

→ Thank you for your comment. As you commented, we added the clinical application information of saxagliptin in revised manuscript.

Round 2

Reviewer 1 Report

Comments and Suggestions for Authors

the authors did most of the required changes and replied in a very professional way in some comments. the paper could be published in the current form 

Reviewer 2 Report

Comments and Suggestions for Authors

please accept